# First-Principles Study for Gas Sensing of Defective SnSe$_2$ Monolayers

**Wei-Ying Cheng [1], Huei-Ru Fuh [2] and Ching-Ray Chang [3],***

[1] Graduate Institute of Applied Physics, National Taiwan University, Taipei 10617, Taiwan; weiying@phys.ntu.edu.tw

[2] Department of Chemical Engineering & Materials Science, Yuan Ze University, Taoyuan 32003, Taiwan; hrfuh@saturn.yzu.edu.tw

[3] Department of Physics, National Taiwan University, Taipei 10617, Taiwan

* Correspondence: crchang@phys.ntu.edu.tw; Tel.: +886-2-3366-5176

**Abstract:** We report the interaction between gas molecules (NO$_2$ and NH$_3$) and the SnSe$_2$ monolayers with vacancy and dopants (O and N) for potential applications as gas sensors. Compared with the gas molecular adsorbed on pristine SnSe$_2$ monolayer, the Se-vacancy SnSe$_2$ monolayer obviously enhances sensitivity to NO$_2$ adsorption. The O-doped SnSe$_2$ monolayer shows similar sensitivity to the pristine SnSe$_2$ monolayer when adsorbing NO$_2$ molecule. However, only the N-doped SnSe$_2$ monolayer represents a visible enhancement for NO$_2$ and NH$_3$ adsorption. This work reveals that the selectivity and sensitivity of SnSe$_2$-based gas sensors could be improved by introducing the vacancy or dopants.

**Keywords:** SnSe$_2$; defect; O-doped; N-doped; gas sensors; first-principle study

## 1. Introduction

Recently, two-dimensional transition metal dichalcogenide (2D-TMD) materials have gained great attention due to their unique structural and electrical properties. Since graphene was introduced into the research, other families of 2D materials with layered structures are also fast emerging for some better applications. 2D-TMD materials with a narrow tunable band gap and replaceable cation and anion [1–3] are more advantageous than the pristine graphene which lacks band gap. SnSe$_2$, a IV–VI semiconductor, has been widely studied for optoelectronic and thermoelectric applications [4,5]. For example, SnSe$_2$ is used as a high-performance photodetector that shows relatively fast photoresponse at room temperature with a high photo-to-dark ratio [4].

2D-TMDs materials have been applied as gas sensors due to their large surface-area-to-bulk ratio. The layered SnX$_2$ (X = S, Se) nanosheets show a significant sensitivity to individual molecules, such as NO$_2$ and NH$_3$ [3,6–8]. The SnX$_2$ (X = S, Se)-based gas sensors show a good response to NO$_2$ at room temperature [6–8]. Furthermore, the SnSe$_2$ monolayer shows a higher sensitivity for NO$_2$ molecule adsorption than SnS$_2$ in our related work [7,8]. Also, the charge transfers and the flat band contributed by gas adsorption induces the conductivity difference of the SnSe$_2$ monolayer reported in our previous study [8].

More recently, the doped SnSe$_2$ nanosheets have been widely studied because of its interesting electronic and optoelectronic properties, [9–13]. Huang et al. [9] systematically studied n-type/p-type and isoelectronic doping cases on SnSe$_2$ nanosheets based on density functional theory (DFT). Huang et al. [9] suggest that P and As are not promising candidates for p-type doping because those atoms contribute trap states near the Fermi level (E$_F$). Although the N atom is a promising candidate for p-type doping which induces states near the valence band maximum (VBM), it is difficult to achieve

the N-doped $SnSe_2$ in reality. For O-, S-, Te-doped $SnSe_2$, the density of states (DOS) of all doped $SnSe_2$ are similar to that of pristine $SnSe_2$ monolayer [9]. On the other hand, n-type doping of F, Cl and Br are highly recommended, especially Br, since the states near both the VBM and the conduction band minimum (CBM) result in a high carrier density and conductivity [10–12]. For the gas sensor, the $SnSe_2$ monolayer demonstrates a high sensitivity and charge transfer to $NO_2$ [8].

The defective graphene would enhance sensitivity for gas sensing which has been theoretically and experimentally reported [14,15]. Zhang et al. [14] theoretically reveal that the defective graphene has stronger interaction with CO, NO and $NO_2$ than the pristine graphene. Also, the B-doped graphene gives the tightest binding with $NH_3$. Lee et al. [15] demonstrate the defect-engineered graphene oxide chemical sensors, which exhibit ultrahigh sensitivity for $NO_2$ and $NH_3$ from the experimental data. However, there is no related study regarding the gas-sensing properties of the doped $SnSe_2$ nanosheets.

In this work, on the basis of DFT, we investigate the gas detection properties for $NO_2$ and $NH_3$ adsorbed on the defective $SnSe_2$ monolayer by substitution of the Se site with a single vacancy, O or N atoms. To the best of our knowledge, the gas sensors of the defective $SnSe_2$ monolayers are investigated for the first time. In order to understand the sensing mechanism, we report adsorption energy, charge transfer, DOS and structural parameters of gas molecules adsorption on defective $SnSe_2$ monolayers. We also discuss and compare the gas-sensing parameters of defective $SnSe_2$ monolayers with the pristine $SnSe_2$ monolayer. We find that the vacancy and doped $SnSe_2$ monolayer can enhance the selectivity and sensitivity of gas sensing.

## 2. Method

The $SnSe_2$ monolayer structure is based on the experimental lattice parameters of bulk $SnSe_2$ [16]. The initial lattice constants of $SnSe_2$ monolayer are $a = b = 3.81$ Å and thickness of vacuum is set about 16 Å. All calculated structures contain the fully lattice constants and atom positions optimization. After structure optimization, the lattice constants of the $SnSe_2$ monolayer are $a = b = 3.87$ Å and the energy gap is 0.78 eV. The Visualization for Electronic and Structural Analysis (VESTA) software is a 3D visualization program for structural models [17]. We use VESTA to show the crystal structure and the defective single layer in this work. The calculation was implemented in the Vienna Ab initio Simulation Package (VASP) and performed by the projector augmented wave (PAW) method with the Perdew–Burke–Ernzerhofer (PBE) generalized gradient approximation (GGA) [18,19]. The $SnSe_2$ monolayer is constructed with a $3 \times 3 \times 1$ supercell in order to perform gas molecules adsorption calculation. The energy cutoff for a plane-wave basis was set up to 400 eV within the $12 \times 12 \times 1$ Monkhorst–Pack $k$-point grid for all study cases. The energy convergence threshold and force convergence criteria were set to $10^{-5}$ eV per unit cell and 0.01 eV Å$^{-1}$.

We also discussed the vacancy and doped $SnSe_2$ monolayers that adsorb $NO_2$ and $NH_3$ gas molecules. First of all, we put the $NO_2$ and $NH_3$ molecules on 3 Å above the Se site or the dopants. We calculated two orientations of N atom, the N atom of gas molecules toward and backward the defective $SnSe_2$ monolayers, named N-bottom and N-top, respectively. In order to understand the defect effect of $SnSe_2$ monolayers adsorbing gas molecule, a Se atom was substituted by a vacancy or a dopant atom (N and O). In this paper, the structures of the $SnSe_2$ monolayers with gas adsorption were relaxed. The initial configurations for gases adsorption on the defective $SnSe_2$ monolayers are illustrated in Figures S1 and S2 of the supplementary material.

## 3. Results

### 3.1. SnSe₂ Monolayers

The pristine 1T phase $SnSe_2$ monolayer is a hexagonal crystal structure as shown in Figure 1a. A Sn atom is sandwiched between two Se atoms which form a Se–Sn–Se arrangement with ABC stacking as shown in Figure 1e. The defective $SnSe_2$ monolayer with Se vacancy in the center of supercell is represented in Figure 1b,f. Figure 1c,g are the defective $SnSe_2$ monolayer in which a Se

atom is substituted by O atom. Furthermore, Figure 1d,h are the defective SnSe$_2$ monolayer in which a Se atom is substituted by a N atom. The O and N dopants are sucked into the vacancy site as shown in Figure 1g,h.

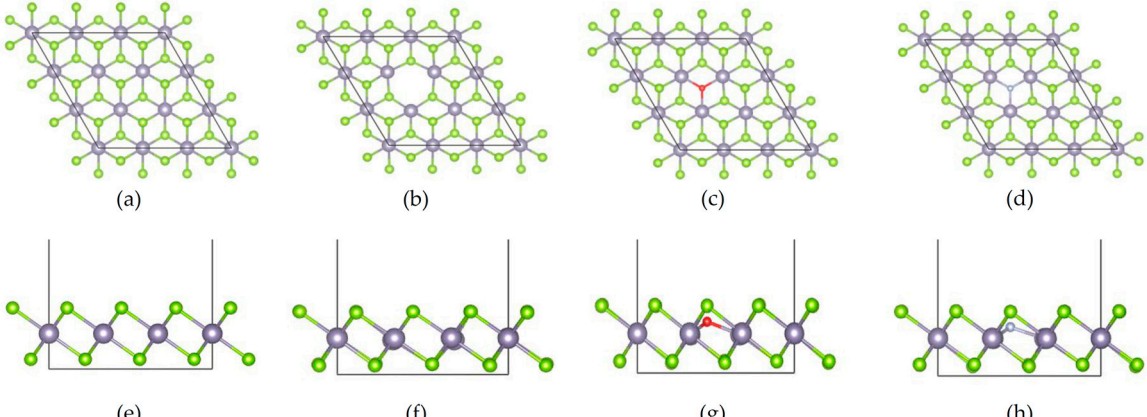

**Figure 1.** The structures of different SnSe$_2$ monolayer systems which including the top view of (**a**) pristine, (**b**) Se vacancy, (**c**) O-doped and (**d**) N-doped SnSe$_2$ monolayer and the side view of (**e**) pristine, (**f**) Se vacancy, (**g**) O-doped and (**h**) N-doped SnSe$_2$ monolayer, respectively. The Sn, Se, O, and N atom are purple, green, red and grey, respectively.

Figure 2 is the DOS with or without defects on the SnSe$_2$ monolayer. The pristine SnSe$_2$ monolayer has an indirect band gap of 0.78 eV as shown in Figure 2a, which is consistent with the previous reported value [8,20]. In the Se-vacancy SnSe$_2$ monolayer, there are occupied states very near E$_F$ as shown in Figure 2b. However, the DOS are very different between the O-doped and the N-doped SnSe$_2$ monolayer. The DOS of the O-doped SnSe$_2$ monolayer is similar to the pristine SnSe$_2$ monolayer with the energy gap 0.84 eV because the states contributed from impurities are far from E$_F$, as shown in Figure 2c. The resultant impurity state from the doped O atom is near the band edge, above 0.8 eV and below −0.1 eV denoted by the orange line in Figure 2c. For the N-doped SnSe$_2$ monolayer, the partial density of state (PDOS) induced by the N dopant is near VBM denoted by the blue line in Figure 2d. The DOS of the SnSe$_2$ monolayer with N and O dopant also is consistent with the result of Huang et al. [9].

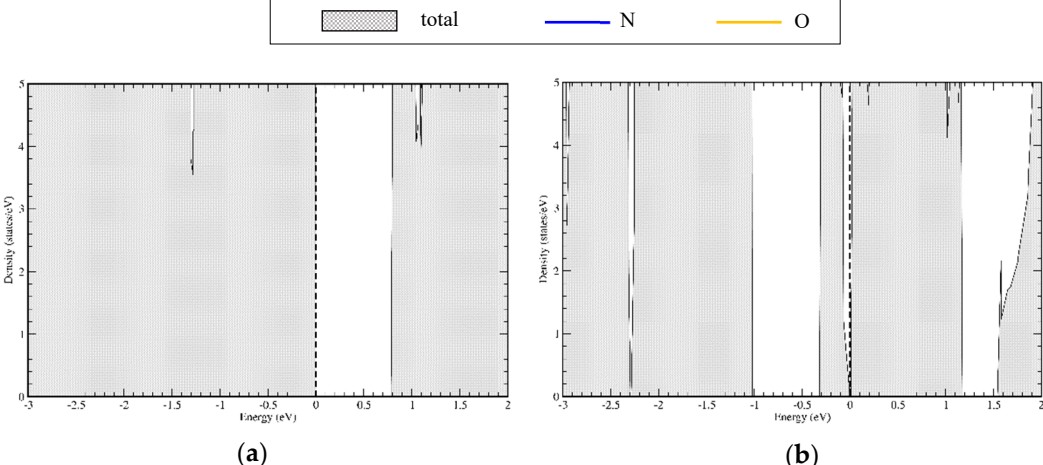

**Figure 2.** *Cont.*

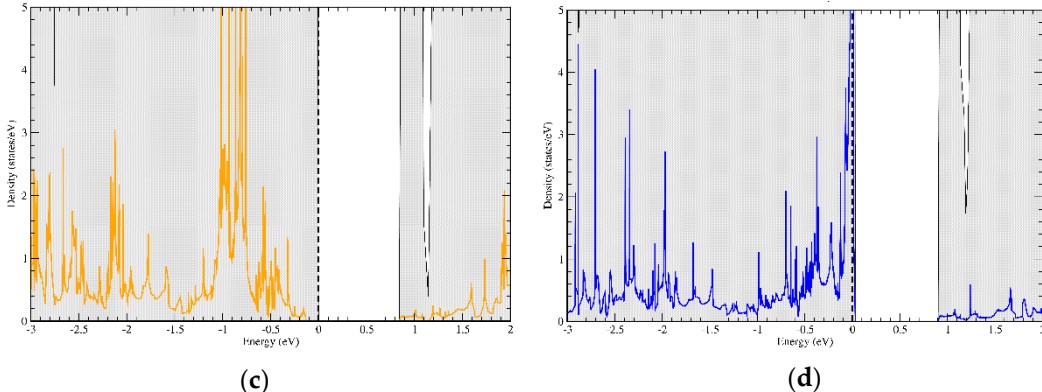

(**c**)  (**d**)

**Figure 2.** The density of states (DOS) of the (**a**) pristine, (**b**) Se vacancy, (**c**) O-doped and (**d**) N-doped SnSe₂ monolayer. The shadowed area is the total DOS. The blue and orange lines are the partial density of state (PDOS) of N and O atoms, respectively. The $E_F$ is denoted by a dashed line and shift to zero.

### 3.2. Gases Adsorbed on Different SnSe₂ Monolayers

In this section, we discuss the most stable configurations of NO₂ and NH₃ molecules adsorbed on different SnSe₂ monolayer systems, whichare listed in Table 1.

**Table 1.** The most stable structural configurations of different gas molecules adsorbed on SnSe₂ monolayer systems. $E_{ad}$, $d_{atom-atom}$, h and $\Delta Q_b$ of molecules adsorbed on the pristine and defective SnSe₂ monolayers.

| Gas | SnSe₂ System | Gas Orientation | $E_{ad}$ (eV) | $d_{atom-atom}$ (Å) | h (Å) | $\Delta Q_b$ | Source |
|---|---|---|---|---|---|---|---|
| NO₂ | Pristine | N-top | −0.29 | 2.97 | 2.45 | −0.164e | Reference [8] |
| | Se vacancy | N-bottom | −1.84 | 2.28 | −0.08 | −0.926e | This work |
| | O-doped | N-top | −0.32 | 3.05 | 1.75 | −0.145e | This work |
| | N-doped | N-bottom | −2.98 | 1.38 | 0.72 | −0.368e | This work |
| NH₃ | Pristine | N-bottom | −0.18 | 3.35 | 2.47 | 0.028e | Reference [8] |
| | Se vacancy | N-bottom | −0.82 | 2.90 | 0.45 | 0.016e | This work |
| | O-doped | N-top | −0.13 | 3.06 | 1.93 | 0.000e | This work |
| | N-doped | N-bottom | −0.36 | 2.46 | 1.45 | 0.215e | This work |

The adsorption energy ($E_{ad}$) of gas molecules on SnSe₂ monolayers is defined as

$$E_{ad} = E_{Gas + SnSe2} - E_{SnSe2} - E_{Gas}, \tag{1}$$

where $E_{Gas + SnSe2}$ is the total energy of the gas molecules adsorbed on SnSe₂ monolayers, $E_{SnSe2}$ is the total energy of the SnSe₂ monolayer and $E_{Gas}$ is the energy of isolated gas molecule. A negative $E_{ad}$ value means that gas molecules on the SnSe₂ monolayer is energetically favorable. In our previous work [8], the adsorption energy of NO₂ adsorbed on the pristine SnSe₂ monolayer is higher than that of NH₃ adsorption in theoretical prediction. In experimental demonstrations, the pristine SnSe₂ monolayer can detect NO₂ in lower concentrations than NH₃. Although the value of adsorption energy is not direct and not the only relation to the sensitivity, both of them could coincide in terms of theoretical calculation and experimental measurements. Generally, the $E_{ad}$ values of NO₂ on SnSe₂ monolayer systems (−0.29 to −2.98 eV) are larger than NH₃ (−0.13 to −0.82 eV). This indicates that the sensitivity of NO₂ is higher than NH₃ on SnSe₂ monolayers. Moreover, the distance between NO₂ and SnSe₂ monolayers is smaller than the distance between NH₃ and SnSe₂ monolayer.

The Bader charge population analysis [21] is summarized in Table 1 where negative $\Delta Q_b$ indicates electron charge transfer from SnSe₂ to a gas molecule and a positive $\Delta Q_b$ shows charge transfer from a gas molecule to SnSe₂ monolayers. Our calculation shows that the NO₂ is an electron charge acceptor,

whereas $NH_3$ is an electron charge donor in all of the study cases. Moreover, the absolute values of $\Delta Q_b$ for $NO_2$ adsorbed on $SnSe_2$ monolayers are greater than that for $NH_3$ adsorbed on $SnSe_2$ monolayers.

In Table 1, we use the two parameters, $d_{atom\text{-}atom}$ and h, to describe the position of gas molecules adsorbed on the $SnSe_2$ monolayers. The $d_{atom\text{-}atom}$ means the shortest distance between the lowest atom of a gas molecule and the highest atom of $SnSe_2$ monolayer. The h indicates the vertical distance between them. We mark $d_{atom\text{-}atom}$ and h in the structure of gas adsorbed on the $SnSe_2$ monolayers to compare the positions of gas adsorption in different $SnSe_2$ monolayers. This would be shown in following figures. Furthermore, we also discuss the h, $E_{ad}$ and $\Delta Q_b$ to analyze the gas-sensing mechanism for defective $SnSe_2$ monolayers as shown in Table 1.

In the following sections, the detail will be discussed about adsorption energy, charge transfer, DOS and structural parameters of gas molecules adsorption on defective $SnSe_2$ monolayers.

### 3.2.1. $NO_2$ Adsorption

Figure 3 shows the optimized structures of $NO_2$ adsorbed on $SnSe_2$ monolayers, including top views as shown in Figure 3a–d and side views, as shown in Figure 3e–h. When $NO_2$ is adsorbed on the pristine $SnSe_2$ monolayer, the site of $NO_2$ adsorption is near the Se atom, the $d_{O\text{-}Se}$ is 2.97 Å and the orientation is N-top, as shown in Figure 3a,e. When $NO_2$ is adsorbed on the Se-vacancy $SnSe_2$ monolayer, the site is near the Sn atom, the $d_{O\text{-}Sn}$ is 2.28 Å and the orientation is N-bottom, as shown in Figure 3b,f. The optimized structure of $NO_2$ on the O-doped $SnSe_2$ monolayer is upon the site of the O atom, the $d_{O\text{-}O}$ is 3.05 Å and the orientation is N-top, as shown in Figure 3c,g. The optimized structure of $NO_2$ on the N-doped $SnSe_2$ monolayer is located on the N site and the $d_{N\text{-}N}$ is 1.38 Å with the orientation N-bottom, as shown in Figure 3d,h. Moreover, the vertical distances h between $NO_2$ and the defective $SnSe_2$ monolayers are 2.45 Å, −0.08 Å, 1.75 Å, 0.72 Å, respectively. The negative value of h means the gas molecule is sucked into the $SnSe_2$ monolayer. In conclusion, the values of h between $NO_2$ and the defective $SnSe_2$ monolayer are shorter than $NO_2$ adsorbed on the pristine $SnSe_2$ monolayer. When comparing with orientation and the h, the structures of $NO_2$ adsorbed on the $SnSe_2$ monolayers could divide into two groups. First, the $NO_2$ adsorption on the pristine and O-doped $SnSe_2$ monolayer are both N-top orientation. Second, the $NO_2$ adsorption on the Se-vacancy and N-doped $SnSe_2$ monolayer are both with N-bottom orientation. The values of h are smaller for $NO_2$ adsorption on the Se-vacancy and N-doped $SnSe_2$ monolayer than those on the pristine and O-doped $SnSe_2$ monolayer.

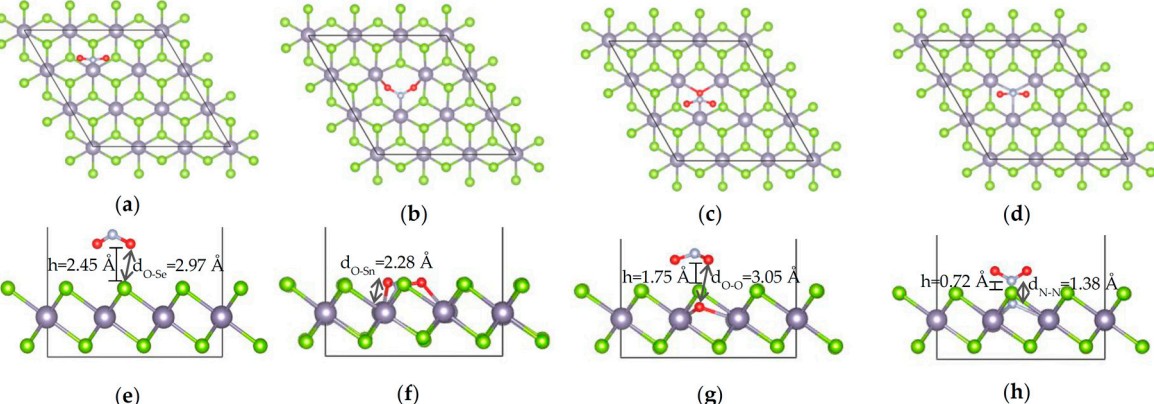

**Figure 3.** The most stable configurations of different $SnSe_2$ monolayers adsorbing $NO_2$ molecule. The top view of (**a**) pristine, (**b**) Se vacancy, (**c**) O-doped and (**d**) N-doped $SnSe_2$ monolayer; the side view of (**e**) pristine, (**f**) Se vacancy, (**g**) O-doped and (**h**) N-doped $SnSe_2$ monolayer, respectively. The Sn, Se, O, and N atom are purple, green, red and grey, respectively.

The value of $E_{ad}$ and $\Delta Q_b$ of $NO_2$ adsorbed on the pristine $SnSe_2$ monolayer are −0.29 eV and −0.164e, respectively [8]. Furthermore, the $E_{ad}$ of $NO_2$ adsorbed on the Se-vacancy and doped $SnSe_2$ monolayer are greater than $NO_2$ adsorbed on the pristine $SnSe_2$ monolayer, as shown in Table 1.

The value of h between $NO_2$ and the Se-vacancy $SnSe_2$ monolayer is −0.08 Å. This induces a large adsorption energy and Bader charge because the $NO_2$ molecule is sucked into the vacancy site. The $E_{ad}$ and $\Delta Q_b$ are up to −1.84 eV and −0.926e, respectively, as listed in Table 1. The value of $E_{ad}$ is greater than the pristine and the O-doped $SnSe_2$ monolayer but smaller than the N-doped $SnSe_2$ system. The value of $\Delta Q_b$ = −0.926e of $NO_2$ adsorbed on the Se-vacancy $SnSe_2$ monolayer is the maximum of that on the $SnSe_2$ monolayers. This result indicates that $NO_2$ molecule would strongly interact with the Se-vacancy $SnSe_2$ monolayer.

As mentioned, the structure of $NO_2$ adsorbed on the O-doped $SnSe_2$ monolayer is similar with on the pristine $SnSe_2$ monolayer despite of the O atom replacing the Se atom. In the detail, the value of h between $NO_2$ and the O-doped $SnSe_2$ monolayer is 1.75 Å, which is also a little smaller than that on the pristine $SnSe_2$ monolayer. Furthermore, the O-doped $SnSe_2$ monolayer adsorbs the $NO_2$ molecule and would induce a little increase in the adsorption energy but decrease the Bader charge by about −0.32 eV and −0.145e, respectively, as shown in Table 1.

When $NO_2$ is adsorbed on the N-doped $SnSe_2$ monolayer, the value of h between $NO_2$ and the N-doped $SnSe_2$ monolayer is 0.72 Å. This indicates that the strong interaction comes from the N-N atoms interaction between $NO_2$ and the N dopant. The $E_{ad}$ = −2.98 eV of $NO_2$ adsorbed on the N-doped $SnSe_2$ monolayer is greater than on the Se-vacancy $SnSe_2$ monolayer. $\Delta Q_b$ = −0.368e of $NO_2$ adsorbed on the N-doped $SnSe_2$ monolayer is smaller than on the Se-vacancy $SnSe_2$, but greater than the pristine and O-doped $SnSe_2$. The relatively high $E_{ad}$ and $\Delta Q_b$ demonstrate strong interaction between $NO_2$ and the N-doped $SnSe_2$ monolayer.

The DOS of $NO_2$ adsorbed on different $SnSe_2$ monolayers is shown in Figure 4. The shadowed area is the total DOS. The blue and orange lines are the PDOS of N and O, respectively. The $E_F$ is denoted by a dashed line. DOS of $NO_2$ adsorption on different $SnSe_2$ monolayers, as shown in Figure 4a–d, are quite different to Figure 2a–d without $NO_2$ adsorption cases. This indicates that all the $SnSe_2$ monolayers could induce obvious difference of electronic structure before/after $NO_2$ adsorption.

When $NO_2$ is adsorbed on the pristine $SnSe_2$, the total DOS presents a trap state across $E_F$ in the energy range about −0.04 eV to −0.02 eV, as shown in Figure 4a. A large amount of PDOS induced by N and O atoms of gas molecule is located in the aforementioned trap state, which induces the flat band and trap electron on it as mentioned in previous work [8]. The pattern of DOS in Figure 4a is different from DOS of the pristine $SnSe_2$ monolayer without adsorption as shown in Figure 2a.

For $NO_2$ adsorption on the Se-vacancy $SnSe_2$ monolayer, the total DOS presents a bandwidth near the $E_F$ with an energy range of −0.19 eV to −0.12 eV, as shown in Figure 4b. A considerable amount of PDOS contributed by N and O atoms of the gas molecule is located the aforementioned bandwidth. Therefore, the pattern of DOS in Figure 4b is quite different from DOS of the Se-vacancy $SnSe_2$ without adsorption as shown in Figure 2b.

When $NO_2$ is adsorbed on the O-doped $SnSe_2$ monolayer, the total DOS presents a trap state across $E_F$ in the energy range about −0.06 eV to −0.03 eV, as shown in Figure 4c. A large amount of PDOS induced by N and O atoms of the gas molecule is located in the aforementioned trap state, which corresponds to a flat band. The trap state would trap electrons on it and decrease the carrier mobility. The DOS of $NO_2$ adsorption the O-doped $SnSe_2$ monolayer is very similar with the pristine $SnSe_2$ monolayer, but different from the O-doped $SnSe_2$ monolayer without adsorption as shown in Figure 2c.

For $NO_2$ adsorption on the N-doped $SnSe_2$ monolayer, the total DOS shown in Figure 4d demonstrates a pattern for a semiconductor with an energy gap of about 0.86 eV. The PDOS introduced by N and O atoms of gas molecule below $E_F$ and above 0.86 eV marked in blue and orange lines as shown in Figure 4d, which is quite different from Figure 4a–c. It is worth noting that the peak of PDOS of the N atom near the $E_F$ shifts to −0.20 eV when $NO_2$ is adsorbed as shown in Figure 4d. It indicates

that PDOS of the N atoms near $E_F$ in Figure 2d are moved to below $E_F$ because the strong interaction occurred after NO$_2$ adsorption.

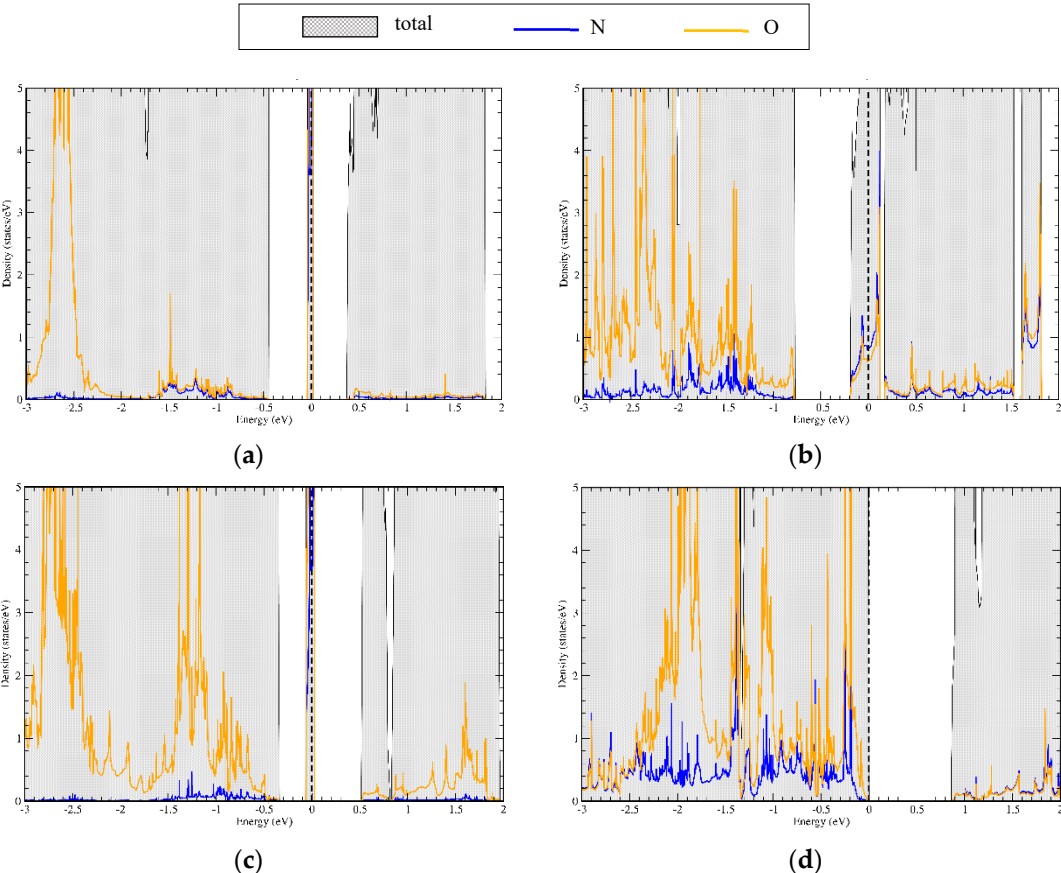

**Figure 4.** DOS of NO$_2$ adsorption on the (**a**) pristine, (**b**) Se-vacancy, (**c**) O-doped and (**d**) N-doped SnSe$_2$ monolayer. The shadowed area is the total DOS. The blue and orange lines are the PDOS of N and O, respectively. The $E_F$ is denoted by a dashed line and shift to zero.

### 3.2.2. NH$_3$

Figure 5 shows the optimized structures of NH$_3$ adsorbed on SnSe$_2$ monolayers, including top views as shown in Figure 5a–d and side views as shown in Figure 5e–h. When NH$_3$ is adsorbed on the pristine SnSe$_2$ monolayer, the site is positioned above the Sn atom, the $d_{N-Se}$ is 3.35 Å and the orientation is N-bottom as shown in Figure 5a,e. When NH$_3$ is adsorbed on the Se-vacancy SnSe$_2$ monolayer, the site is positioned above the site of the single vacancy, the $d_{H-Se}$ is 2.90 Å and the orientation is N-bottom, as shown in the Figure 5b,f. The most stable configuration of NH$_3$ adsorbed on the O-doped SnSe$_2$ monolayer is positioned at the O site, the $d_{H-O}$ is 3.06 Å and the orientation is N-top as shown in Figure 5c,g. The optimized structure of NH$_3$ adsorbed on the N-doped SnSe$_2$ monolayer is positioned on the N site, the $d_{N-N}$ is 2.46 Å and the orientation is N-bottom as shown in Figure 5d,h. It is worth noting that the orientation of NH$_3$ adsorbed on O-doped SnSe$_2$ monolayer is different from the other SnSe$_2$ monolayers, which is N-top orientation but others are N-bottom. The vertical distances h between NH$_3$ and the different SnSe$_2$ monolayers are 2.47 Å, 0.45 Å, 1.93 Å, 1.45 Å as shown in Figure 5e–h, respectively. The values of h of NH$_3$ adsorbed on the Se-vacancy and doped SnSe$_2$ monolayer are smaller than NH$_3$ adsorbed on the pristine SnSe$_2$ monolayer.

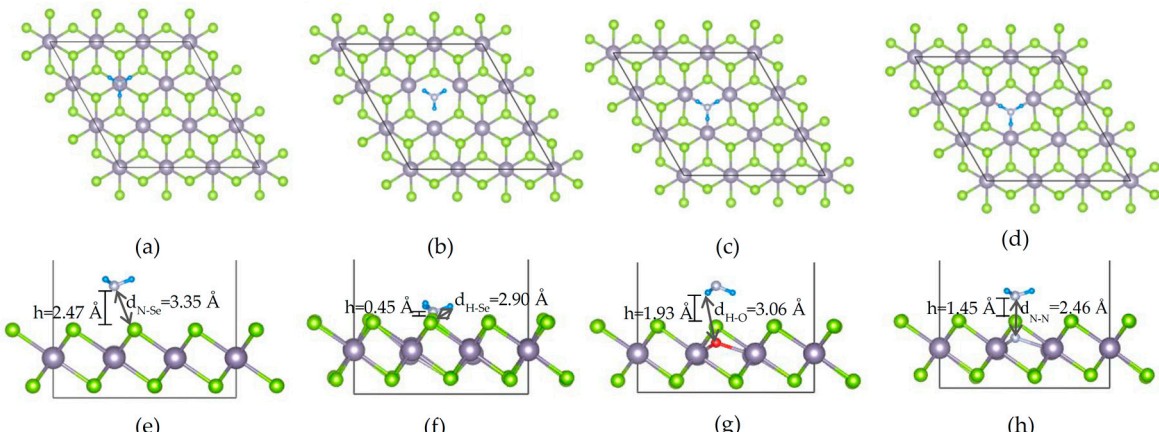

**Figure 5.** The most stable configurations of different $SnSe_2$ monolayers adsorbing $NH_3$ molecule. The top view of (**a**) pristine, (**b**) Se vacancy, (**c**) O-doped and (**d**) N-doped $SnSe_2$ monolayer; the side view of (**e**) pristine, (**f**) Se vacancy, (**g**) O-doped and (**h**) N-doped $SnSe_2$ monolayer, respectively. The Sn, Se, O, N and H atoms are purple, green, red, grey and cyan, respectively.

When $NH_3$ is adsorbed on the pristine $SnSe_2$ monolayer, the value of h, $E_{ad}$ and $\Delta Q_b$ are 2.47 Å, −0.18 eV and 0.028e [8], respectively. Compared to the value of $E_{ad}$ and h in the Table 1, the $E_{ad}$ of $NH_3$ adsorption almost follows a positive correlation to h, except for $NH_3$ adsorption on the O-doped $SnSe_2$ monolayer.

When $NH_3$ is adsorbed on the Se-vacancy $SnSe_2$ monolayer, the value of h = 0.45 Å is the minimum among $NH_3$ adsorption on $SnSe_2$ monolayers. Furthermore, the value of $E_{ad}$ = −0.82 eV of $NH_3$ on the Se-vacancy $SnSe_2$ monolayer reaches the maximum compared with other cases. However, the charge transfer amount $\Delta Q_b$ = 0.016e of $NH_3$ adsorbed on the Se-vacancy $SnSe_2$ monolayer is smaller than gas molecule on the pristine $SnSe_2$ (0.028e).

For $NH_3$ adsorption on the O-doped $SnSe_2$ monolayer, the value of h = 1.93 Å between $NH_3$ and the O-doped $SnSe_2$ monolayer is smaller than that on the pristine $SnSe_2$ monolayer. The values of $E_{ad}$ and $\Delta Q_b$ of $NH_3$ on the O-doped $SnSe_2$ monolayer are −0.13 eV and 0.000e, which are both the minimum among the $NH_3$ adsorption. This indicates that the ability of $NH_3$ adsorption of the O-doped $SnSe_2$ monolayer is weaker than the pristine $SnSe_2$ monolayer.

For $NH_3$ adsorption on the N-doped $SnS_2$ monolayer, the value of h = 1.45 Å between $NH_3$ and the O-doped $SnSe_2$ monolayer is smaller than that on the pristine $SnSe_2$ monolayer. $E_{ad}$ also has a greater value −0.36 eV than that on the pristine $SnSe_2$ monolayer, and $\Delta Q_b$ of $NH_3$ on the N-doped $SnSe_2$ monolayer has a maximum value 0.215e. This shows an enhancement of $NH_3$ adsorption on the N-doped $SnSe_2$ monolayer.

Figure 6 shows DOS of $NH_3$ adsorption on different $SnSe_2$ monolayers. The blue, orange, cyan lines are the PDOS of N, O, H, respectively. The DOS of $NH_3$ adsorption on the pristine $SnSe_2$ monolayers demonstrates a pattern of a semiconductor with energy gap as shown in Figure 6a. However, the DOS of $NH_3$ adsorption on the defective $SnSe_2$ monolayers presents a flat band as shown in Figure 6b–d.

When $NH_3$ adsorbed on the pristine $SnSe_2$ monolayer, the total DOS shown in Figure 6a demonstrates a pattern for a semiconductor with an energy gap about 0.81 eV. The PDOS introduced by N and H atoms of gas molecules below $E_F$ and above 0.81 eV is marked in blue and cyan lines as shown in Figure 6a. There is no obviously change of the electronic structure of the pristine $SnSe_2$ as shown in Figure 2a, which is consistent with our previous work [8].

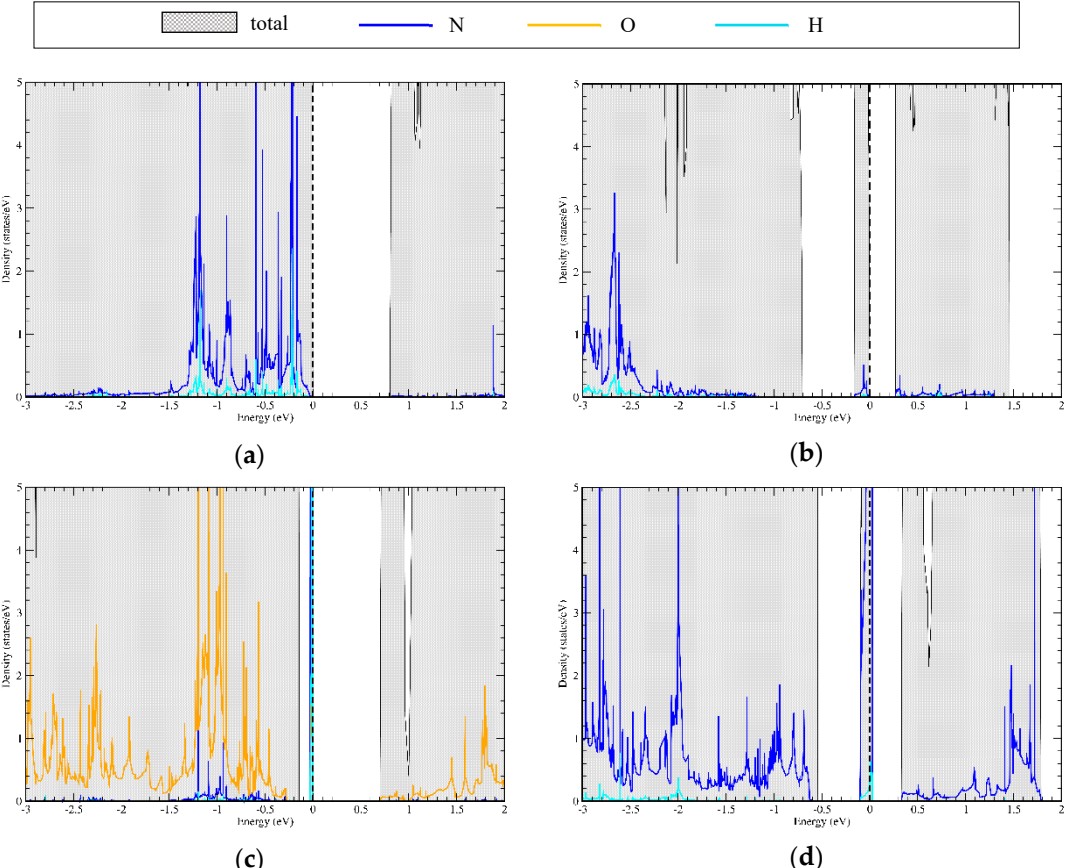

**Figure 6.** DOS of $NH_3$ adsorption on the (**a**)pristine, (**b**)Se vacancy, (**c**)O-doped and (**d**)N-doped $SnSe_2$ monolayer. The shadowed area is the total DOS. The blue, orange, cyan lines are the PDOS of N, O, H, respectively. The $E_F$ is denoted by a dashed line and shift to zero.

When $NH_3$ adsorbed on the Se-vacancy $SnSe_2$ monolayer, the total DOS presents a bandwidth in energy range of −0.17 eV to $E_F$ as shown in Figure 6b. The DOS pattern of Figure 6b is a semiconductor with energy gap about 0.27 eV. Only a small amount of PDOS contributed by N and H atoms of gas molecule is located the aforementioned bandwidth. This is so that $NH_3$ adsorption does not induce obvious difference on the DOS of the Se-vacancy $SnSe_2$ without adsorption as shown in Figure 2b.

For the O-doped $SnSe_2$ monolayer with $NH_3$ adsorption, the total DOS presents a flat band below $E_F$ as shown in Figure 6c. A large amount of PDOS contributed by N and H atoms of gas molecules is located in the aforementioned flat band denoted by blue and cyan lines in Figure 6c. Because the flat band is located below $E_F$, the DOS pattern of Figure 6c is a semiconductor with energy gap about 0.70 eV. Therefore, the $NH_3$ adsorption does not change the pattern of DOS of the O-doped $SnSe_2$ monolayer as shown in Figure 2c.

For the N-doped $SnSe_2$ monolayer with $NH_3$ adsorption, the total DOS presents a trap state across $E_F$ in the energy range about −0.10eV to −0.03eV as shown in Figure 6d. A large amount of PDOS induced by N and H atoms of gas molecules is located in the aforementioned trap state. The trap state responds to a flat band which would decrease the mobility of the $SnSe_2$ systems after gas adsorption. The pattern of DOS in Figure 6d is different from that of the N-doped $SnSe_2$ monolayer without adsorption as shown in Figure 2d.

Comparing the DOS of $NH_3$ adsorption on different $SnSe_2$ monolayers as shown in Figure 6a–d with the $SnSe_2$ monolayers without adsorption as shown in Figure 2a–d, only the N-doped $SnSe_2$ monolayer could induce obvious differences of electronic structure before/after $NH_3$ adsorption.

## 4. Discussion

We discuss the gas-sensing parameters, including $E_{ad}$ and $\Delta Q_b$, compared to gas molecules adsorbed on the pristine $SnSe_2$ monolayer and have demonostrated the high selectivity and sensitivity of the defective and doped $SnSe_2$ monolayer for gas-sensor candidates. The alteration of electronic structure is reflected by the change in electrical conductance of the $SnSe_2$ monolayers. We also discuss the DOS of gas adsorptions in the $SnSe_2$ monolayers to illustrate the change in the electrical conductance due to alteration of the electronic structure.

### 4.1. Gas Molecules on the Se-Vacancy SnSe₂ Monolayer

When $NO_2$ adsorbed on the Se-vacancy $SnSe_2$ monolayer, the values of $E_{ad} = -1.84$ eV and $\Delta Q_b = -0.926e$ were the maximum among that of the $SnSe_2$ monolayers. This reveals strong interaction between $NO_2$ molecule and the Se-vacancy $SnSe_2$ monolayer. For the Se-vacancy $SnSe_2$ monolayer, $NO_2$ adsorption induces wide-ranging trap states crossover $E_F$ to decrease the carrier mobility as represented by the pattern of DOS in Figure 4b.

However, the $NH_3$ molecule is adsorbed on the Se-vacancy $SnSe_2$ monolayer with higher $E_{ad} = -0.82$ eV and lower $\Delta Q_b = 0.016e$ compared to the pristine $SnSe_2$ monolayer ($E_{ad} = -0.18$ eV, $\Delta Q_b = 0.028e$ [8]). $NH_3$ adsorption on the Se-vacancy $SnSe_2$ monolayer contributes DOS below $E_F$, as shown in Figure 6b. However, the charge transfer amount of $NH_3$ on the Se-vacancy $SnSe_2$ monolayer is smaller than that of $NH_3$ on the pristine $SnSe_2$ monolayer. Therefore, we cannot be sure of the conductivity difference of the Se-vacancy $SnSe_2$ monolayer with/without $NH_3$ adsorption.

The Se-vacancy $SnSe_2$ monolayer shows excellent sensitivity for $NO_2$ molecules, whereas the Se-vacancy $SnSe_2$ monolayer cannot show obvious conductivity difference for $NH_3$ adsorption. In brief, the Se-vacancy $SnSe_2$ monolayer shows enhancement only for $NO_2$.

### 4.2. Gas Molecules on the O-Doped SnSe₂ Monolayer

When $NO_2$ adsorbed on the O-doped $SnSe_2$ monolayer, the values of $E_{ad} = -0.32$ eV and $\Delta Q_b = -0.145e$ are close to the molecule on the pristine $SnSe_2$ monolayer ($E_{ad} = -0.29$ eV, $\Delta Q_b = -0.164e$ [8]). The DOS of the O-doped $SnSe_2$ monolayer is similar to that of the pristine $SnSe_2$ monolayer, so that the similar appearance of DOS occurs in $NO_2$ adsorption on the pristine and O-doped $SnSe_2$ monolayer as shown in Figure 4a,c. The DOS of $NO_2$ on the O-doped $SnSe_2$ monolayer induces trap state crossover $E_F$ and decreases the carrier mobility. The electronic structure altered by $NO_2$ adsorbed on the O-doped $SnSe_2$ monolayer would contribute a change of electrical conductance, just like $NO_2$ adsorption on the pristine $SnSe_2$ monolayer [8].

On the other hand, the value of $E_{ad} = -0.13$ eV and $\Delta Q_b = 0.000e$ for $NH_3$ on the O-doped $SnSe_2$ monolayer are both the minimum among the $NH_3$ adsorption. The DOS of $NH_3$ on the O-doped $SnSe_2$ monolayer is below the $E_F$, which results in similar electrical properties with the O-doped $SnSe_2$ monolayer without gas adsorption. This means that there is no obvious change of electrical conductance.

In conclusion, gas molecules on the O-doped $SnSe_2$ monolayer shows high sensitivity for $NO_2$ adsorption and even with a weaker detection for $NH_3$, compared to gases on the pristine $SnSe_2$. This result indicates that the O-doped $SnSe_2$ monolayer has better selectivity to these two gases in comparison with pristine $SnSe_2$.

### 4.3. Gas Molecules on the N-Doped SnSe₂ Monolayer

When $NO_2$ adsorbed on the N-doped $SnSe_2$ monolayer, $E_{ad} = -2.98$ eV is the maximum among the $NO_2$ adsorption and $\Delta Q_b = -0.368e$ has relatively high value. It indicates that the strong interaction occurred between $NO_2$ molecule and the N-doped $SnSe_2$ monolayer. Moreover, the DOS of the N-doped $SnSe_2$ monolayer before/after $NO_2$ adsorption are totally different, as shown in Figures 2d and 4d.

Compared to the DOS of the N-doped SnSe$_2$ monolayer before/after NO$_2$ adsorption, the DOS crossover E$_F$ of the N-doped SnSe$_2$ moves to below the E$_F$.

When NH$_3$ is adsorbed on the N-doped SnSe$_2$ monolayer, E$_{ad}$ = −0.36 eV also has a greater value than that on the pristine SnSe$_2$ monolayer and $\Delta Q_b$ = 0.215e has a maximum value. In the DOS of NH$_3$ adsorbed on the N-doped SnSe$_2$ monolayer, the trap state is induced to decrease the carrier mobility. This implies that there would be an obvious change of electrical conductance. The N-doped atom is high sensitivity gas sensor for NO$_2$ and NH$_3$ shown as in our DFT calculation.

When setting gas molecules on the N-doped SnSe$_2$, it reveals obvious enhancement for both NO$_2$ and NH$_3$ adsorption.

## 5. Conclusions

In summary, the adsorption of NO$_2$ and NH$_3$ on the Se-vacancy, O-doped and N-doped SnSe$_2$ monolayer are investigated and compared to the pristine SnSe$_2$ monolayer. Due to the high adsorption energy and large charge transfer of gas adsorption on the Se-vacancy SnSe$_2$ monolayer, the Se-vacancy SnSe$_2$ monolayer shows a better sensitivity only to NO$_2$. However, the sensitivity of NH$_3$ adsorbed on the Se-vacancy SnSe$_2$ monolayer has higher adsorption energy but lower charge transfer amount than the pristine SnSe$_2$ monolayer. Furthermore, the O-doped SnSe$_2$ monolayer has similar interaction with NO$_2$ with the pristine SnSe$_2$ monolayer, but weaker interaction with NH$_3$ than the pristine SnSe$_2$ monolayer. This indicates that the O-doped SnSe$_2$ monolayer has similar sensitivity to the pristine SnSe$_2$ monolayer and better selectivity than the pristine SnSe$_2$ monolayer. The N-doped SnSe$_2$ strongly interacts both with NO$_2$ and NH$_3$ and shows obvious sensing enhancement for those two gases. In brief, the vacancy and doped SnSe$_2$ monolayers can enhance the selectivity and sensitivity of gas sensing for NO$_2$ and NH$_3$ molecules. This work demonstrates the potential of the SnSe$_2$-based gas sensors by introducing defects and dopants in the SnSe$_2$ monolayer.

**Supplementary Materials:** The following are available online at http://www.mdpi.com/2076-3417/10/5/1623/s1: Figure S1: The initial configurations of the defective SnSe$_2$ monolayers adsorbing NO$_2$ molecule with N-top and N-bottom orientations, Figure S2: The initial configurations of the defective SnSe$_2$ monolayers adsorbing NH$_3$ molecule with N-top and N-bottom orientations.

**Author Contributions:** W.-Y.C.: Data curation, Data analysis, writing – original draft & review & editing; H.-R.F.: Conceptualization, writing – review & editing; C.-R.C.: Supervision, Writing – review & editing. All authors have read and agreed to the published version of the manuscript.

**Funding:** This research was funded by Ministry of Science and Technology of R. O. C. under grant No. MOST 107-2112-M-002-013-MY3.

**Acknowledgments:** C. R. Chang thanks for supports from Ministry of Science and Technology of R. O. C. under grant No. MOST 107-2112-M-002-013-MY3. H.R. Fuh thanks for supports from Ministry of Science and Technology of R. O. C. under grant No. MOST 107-2112-M-155-001-MY3. W. Y. Cheng and H. R. Fuh thanks the computer and information networking center in National Taiwan University, Taiwan for the computational support.

**Conflicts of Interest:** The authors declare no conflict of interest.

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
