# Peer review of "First-Principles Study for Gas Sensing of Defective SnSe2 Monolayers"

_applsci, doi:10.3390/app10051623_

Round 1

Reviewer 1 Report

The authors have studied the theory of interaction of gaseous molecules on SnSe2 monolayers. Though I personally would not call it "gas sensing" in the title, the topic is potentially interesting to readers in the community. Before publication, however, I recommend a thorough check of English grammar. Furthermore I cannot judge Figs 2 and 6 because the pdf do not contain them.

Author Response

Response to reviewers’ comments

Dear Referees:

We would like to thank you very much for your efforts and time taken to review our work. Your concerns and comments about the paper are much appreciated. In the revised manuscript (marked-up copy as Supporting Information for Review Only), all changes and revisions have been highlighted by formatting the text in a Red color. For your convenience, we also include below, a point-by-point list of the corrections made in response to all your concerns.

------------------------------

REVIEWER REPORT:

------------------------------

Reviewer: 1

Comments and Suggestions for Authors:

The authors have studied the theory of interaction of gaseous molecules on SnSe2 monolayers. Though I personally would not call it "gas sensing" in the title, the topic is potentially interesting to readers in the community. Before publication, however, I recommend a thorough check of English grammar. Furthermore I cannot judge Figs 2 and 6 because the pdf do not contain them.  

Response:

It’s glad to hear that you think the topic is potentially interesting to readers. Gas sensing seems to contain more phenomenon than gas sensor for defective SnSe2 monolayers, which is presented in this paper. And the English grammar is corrected in the revision. Thanks for your suggestion.

Furthermore, Figure 2 and 6 are included in the attached file. I check all figures included in the revision pdf file.  If you still cannot read the figures, please feel free to contact me.

Reviewer 2 Report

This manuscript by Cheng and colleagues reports on DFT studies of the effect of interactions between NO2 and NH3 molecules with defective SnSe2 monolayers. The work is probably publishable, but requires revision.

A basic question is why do the authors study sensing of NH3 and NO2, and only consider those two molecules? I do not see that this is justified. Other atmospheric gasses such as N2, O2 and CO2 should be added if one wants to assess the suitability of using these systems as sensors in any application that I can think of.

I do not think that the changes to the calculated electronic densities of states are well described in the text. For example, were it not for the shift in the *calculated* Fermi energy, there is little practical difference between e.g. Figs 2a and 2b. These types of calculations do not calculate effective Fermi energies well, simply reporting the energy of the highest occupied band for an electrically neutral cell. That's usually fine for bulk systems. But this is a small single layer system, whose effective Fermi level and hence net charge are often highly dependent on the electronic environment they are put in, depending on mounting, electrodes, etc. So to say that the band gap drops from close to an eV to meV scale is misleading. That ~eV gap is still there. There are states above it that are occupied in the calculation because of the local charge imbalance of the defective structure. I do not view it as particularly significant that the region of low state density between particular bond types evident above 1 eV in Fig. 1a happens to drop to no states calculated in Fig. 2b. I suspect that this "gap about 0.0067eV" is highly dependent on k sampling, atomic potentials, etc., and anyway not what will dominate the conductive response of a real system.

The paper does not clearly distinguish between competitive binding, the electronic/electric effect of binding, and how these might affect sensor selectivity and sensitivity. For example, on page 4 it is stated that NO2 adsorption energies are larger than those for NH3. It is then concluded that "the sensitivity of NO2 is higher than NH3 on SnSe2 monolayers." This is not true. The correct conclusion is that in the presence of NO2 and NH3, NO2 will preferentially bind. It says nothing about the sensitivity. For information about sensitivity the electronic response to binding is the key criterion, merely moderated by binding.

I also do not know how the authors could conclude that Se-vacant monolayers do not indicate conductivity differences for NH3 adsorption (page 10). The band gap that the authors focus on changes from meV scale to ~0.3 eV on adsorption. That should give a huge conductivity change.

More minor points:

Line 69: "The VESTA is a 3D visualization program for structural models [17]." True, but irrelevant.

Section 2 does not clarify that structures are relaxed after adsorbate molecules are added.

For the scale of energies involved and the actual precision to which one can calculate these quantities, I would have thought eV would be a better unit than meV. Quoting energies to 1 meV is probably not justified.

I don't see the integration radii mentioned. These are important for the reproducibility of the indicated PDOS.

It is not appropriate to talk about the "crystal structure" of a defective single layer with an adsorbate. It's not a crystal.

Why are the PDOSs of the defect and adsorbate N atoms not separated in Figs 4 and 6?

Line 305: The wrong energy is quoted.

The manuscript requires proof reading for English usage.

Round 2

Reviewer 1 Report

Dear authors,

thank you for your thorough revision of the manuscript. I think it can now be published.